# Comparison of the Fecal Bacteriome of HIV-Positive and HIV-Negative Older Adults

**DOI:** 10.3390/biomedicines11082305

**Published:** 2023-08-19

**Authors:** Matilde Sánchez-Conde, Claudio Alba, Irma Castro, Fernando Dronda, Margarita Ramírez, Rebeca Arroyo, Santiago Moreno, Juan Miguel Rodríguez, Fátima Brañas

**Affiliations:** 1Infectious Diseases Department, Hospital Universitario Ramón y Cajal, Instituto Ramón y Cajal de Investigación Sanitaria (IRYCIS), 28034 Madrid, Spain; fdronda.hrc@salud.madrid.org (F.D.); smguillen@salud.madrid.org (S.M.); 2CIBER de Enfermedades Infecciosas (CIBERINFECT), Instituto de Salud Carlos III, 28220 Madrid, Spain; 3Department of Nutrition and Food Science, Complutense University of Madrid, 28040 Madrid, Spain; c.alba@ucm.es (C.A.); irmacastro@ucm.es (I.C.); jmrodrig@ucm.es (J.M.R.);; 4Infectious Diseases Department, Hospital General Universitario Gregorio Marañón, 28007 Madrid, Spain; ramirezmarga@yahoo.es; 5Geriatric Department, Hospital Universitario Infanta Leonor, 28031 Madrid, Spain

**Keywords:** gut microbiome, HIV, aging, frailty, dysbiosis, depression

## Abstract

HIV infection is considered a scenario of accelerated aging. Previous studies have suggested a link between aging, frailty, and gut dysbiosis, but there is a knowledge gap regarding the HIV population. Our objective was to compare the fecal bacteriome of older people with HIV (PWH) and non-HIV controls, and to assess potential links between gut dysbiosis and frailty. A total of 36 fecal samples (24 from PWH and 12 from non-HIV controls) were submitted to a metataxonomic analysis targeting the V3–V4 hypervariable region of the 16S rRNA gene. High-quality reads were assembled and classified into operational taxonomic units. Alpha diversity, assessed using the Shannon index, was higher in the control group than in the HIV group (*p* < 0.05). The relative abundance of the genus *Blautia* was higher in the HIV group (*p* < 0.001). The presence of *Blautia* was also higher in PWH with depression (*p* = 0.004), whereas the opposite was observed for the genus *Bifidobacterium* (*p* = 0.004). Our study shows shifts in the composition of the PWH bacteriome when compared to that of healthy controls. To our knowledge, this is the first study suggesting a potential link between depression and gut dysbiosis in the HIV population.

## 1. Introduction

HIV infection has been postulated to be a model of accelerated aging due to the chronic activation of the immune system, even when a patient is undergoing optimal immuno-virological control treatment. This immune activation originates from prolonged antigenic stimulation, concurrent infections, and a continuous translocation of gut-associated microbes [1]. Microbial translocation is favored by the HIV-associated impairment of the gut epithelial barrier and the alteration of the gut microbiome [2,3]. In turn, gut dysbiosis contributes to inflammation [4], further disrupting the epithelial barrier, exacerbating microbial translocation, and facilitating the progression of HIV infection [5]. The clinical expression is an increased prevalence of age-related non-HIV-associated comorbidities, including geriatric syndromes, and a rising prevalence of frailty occurring earlier than in the general population [6,7]. 

Frailty is closely associated with a worse clinical prognosis (e.g., falls, morbidity, or death), but there is the possibility of a successful outcome if it is detected at an early stage of the infection [8,9]. Among community-dwelling adults older than 65 years, the prevalence of frailty is approximately 7% [10,11,12,13], while among people with HIV (PWH), it has been reported to be up to twice that for uninfected individuals who are 10 years older [14,15]. In fact, frailty is a clinical marker of age acceleration in both HIV and non-HIV populations [16,17,18]. Gut dysbiosis has been associated with inflammation in both older people and PWH [19], but there is a lack of information regarding older PWH in this field, which constitutes a knowledge gap [20,21,22]. 

As older PWH have an increased risk of frailty, a reversible condition with prognostic value, and may exhibit increased gut dysbiosis, our intention in this pilot study was to compare the composition of the fecal microbiome of PWH and that of non-HIV controls, as well as to assess if there are potential links with frailty in this population.

## 2. Materials and Methods

### 2.1. Study Design and Patients

A total of 36 fecal samples were analyzed in this work. Among them, 24 had been obtained from virologically suppressed PWH (>55 years old) from our frailty cross-sectional study [10], including 7 frail, 9 pre-frail, and 8 robust ones, matched for age and nadir CD4. The samples had been stored frozen (–80 °C) since then. In the frame of such a study, the population was screened for the prevalence of frailty and tested for physical function, depression, nutritional status, and associated factors [10]. In the present study, 12 healthy non-HIV people with a similar age distribution were also included as controls. The ethics committees of each hospital approved the study. All patients and controls signed informed consent forms.

### 2.2. Data Collection

Sociodemographic data, comorbidities (i.e., self-reported and physician-diagnosed chronic conditions), medications (i.e., polypharmacy was defined as taking ≥6 medications), and variables related to HIV infection (i.e., risk practice for HIV infection, the baseline and current immunovirologic status, and the stage of HIV infection at diagnosis) were recorded. Depression status was evaluated using the Short Geriatric Depression Scale (i.e., S-GDS or Yesavage test) [23]. Physical function was assessed by quantifying it using the Short Physical Performance Battery (SPPB) [24]. The recorded laboratory data included HIV-related data (i.e., HIV-RNA, nadir and current CD4 count, and CD4/CD8 rate).

### 2.3. Frailty

Frailty was assessed according to Fried’s frailty phenotype, defined by 5 functional criteria [25], namely, shrinking (unintentional weight loss of ≥4.5 kg or ≥5% of body weight during the previous year), weakness (grip strength adjusted for gender and BMI), poor endurance and energy (self-reported exhaustion identified by 2 questions from the Center for Epidemiologic Studies Depression scale), slowness (based on the time to walk 4 m, adjusting for gender and standing height), and low physical activity level (<383 kcal/week in men and <270 kcal/week in women using the Minnesota Leisure Time Activity Questionnaire). Patients were considered frail when they met at least 3 of the 5 criteria, pre-frail when they met 1 or 2 criteria, and robust when they met no criteria.

### 2.4. Metataxonomic Analysis

A dual-barcoded 2-step PCR reaction was conducted to amplify a fragment of the V3–V4 hypervariable region of the bacterial 16S ribosomal RNA (rRNA) gene. Equimolar concentrations of the universal primers S-D-Bact-0341-b-S-17 (ACACTGACGACATGGTTCTACACCTACGGGNGGCWGCAG) and S-D-Bact-0785-a-A-21 (TACGGTAGCAGAGACTTGGTCTGACTACHVGGGTATCTAATCC) were used. Barcodes used for Illumina sequencing were appended to the 3′ and 5′ terminal ends of the PCR amplicons to allow for the separation of forward and reverse sequences. A bioanalyzer (2100 Bioanalyzer, Agilent, Santa Clara, CA, USA) was used to determine the concentration of each sample. Barcoded PCR products from all samples were pooled at approximately equimolar DNA concentrations and run on a preparative agarose gel. The correctly sized band was excised and purified using a QIAEX II Gel Extraction Kit (Qiagen, Hilden, Germany) and then quantified with PicoGreen (BMG Labtech, Jena, Germany). Finally, 1 aliquot of pooled, purified, and barcoded DNA amplicons was sequenced using the Illumina MiSeq pair-end protocol (Illumina Inc., San Diego, CA, USA) at the facilities of the Scientific Park of Madrid (Spain). The sequences analyzed for this study are available in the BioSample database of the National Center for Biotechnology Information. 

The amplified fragments and results were taxonomically analyzed using the Illumina™ software according to the manufacturer’s guidelines and pipelines (version 2.6.2.3). The resulting high-quality reads were assembled and classified taxonomically into operational taxonomic units (OTUs) by comparison with the Illumina™ software according to the manufacturer’s guidelines and pipelines (version 2.6.2.3) using a Bayesian classification method and a level of similarity of at least 97%.

The concentration of DNA in the 3 blank preparations was approximately 0.01 ng/µL. The decontam R package was used to identify, visualize, and remove contaminating DNA based on the DNA concentration in each sample.

### 2.5. Statistical Analysis

We used descriptive statistics to examine participant characteristics, which were expressed as frequency (percent) for categorical variables, mean (±SD) for normally distributed continuous variables, or median (interquartile range) for continuous variables with a skewed distribution.

We compared continuous variables using the t-test for independent variables. Then, we used the Wilcoxon and Mann–Whitney tests for variables with 2 factors and a non-normal distribution or when the group size was small, and the Kruskal–Wallis test was used with variables with 3 or more factors and a non-normal distribution. We assessed the association between qualitative variables using the chi-square test or the Fisher exact test when the groups were very small. In addition, we used linear regression to assess the differences in biological age between frail and robust patients, and we considered differences to be significant when *p* ≤ 0.05. We used SPSS statistical package (version 20.0).

For bacteriome analysis, quantitative data were expressed as the median and interquartile range (IQR). We assessed differences between groups using Kruskal–Wallis tests and pairwise Wilcoxon rank sum tests to calculate comparisons between groups. Also, we made Bonferroni corrections to control multiple comparisons. We generated a table of amplicon sequence variants’ OTU counts per sample and normalized the bacterial taxon abundances to the total number of sequences in each sample. Then, we studied alpha diversity using the Shannon diversity index with the R vegan package (version 2.5.6). 

We used principal coordinate analysis (PCoA) to evaluate beta diversity and to plot patterns of bacterial community diversity through a distance matrix containing a dissimilarity value for each pairwise sample comparison. We performed quantitative (relative abundance) and qualitative (presence/absence) analyses using the Bray–Curtis index and binary Jaccard index, respectively. Then, we performed an analysis of variance of the distance matrices using the “nonparametric manova test” (PERMANOVA) adonis with 999 permutations, as implemented in the R vegan package, to reveal statistical significance. For multilevel pairwise adonis comparisons, we used the Holm–Bonferroni method for *p*-value correction using the “pairwiseAdonis” R package (version 0.0.1). We performed the linear discriminant analysis (LDA) and effect size (LEfSe) algorithms to predict those taxa that violate the null hypothesis of no difference between the control and PWH groups of patients. We performed this analysis with the online interface Galaxy [26]. 

## 3. Results

The main characteristics of the PWH patients are shown in Table 1.

The sequencing of the 36 fecal samples yielded 5,829,212 high-quality reads (median = 168,180 reads/sample, ranging from 106,093 to 194,180), and, among them, a total of 347 OTUs were detected.

Initially, assessment of the alpha diversity using the Shannon index (median [IQR]) at the OTU level showed statistically significant differences between the PWH and the control groups. More specifically, the diversity of the latter group was significantly higher than the one found for the PWH group (3.74 [3.65–3.94] and 3.56 [3.32–3.69], respectively; *p* < 0.05).

Next, both groups were compared for beta diversity. At the OTU level, the PCoA plots of the Bray–Curtis distance matrix (relative abundance) revealed that most of the samples clustered according to the HIV status (*p* = 0.012; PERMANOVA test; Figure 1A). Similarly, the analysis according to the presence/absence of OTUs (binary Jaccard distance matrix) also revealed the existence of significant differences between both groups (*p* = 0.010; Figure 1B). 

A total of 16 phyla were observed in the fecal samples, with Firmicutes/Bacillota, Bacteroidetes/Bacteroidota, Proteobacteria/Pseudomonadota, and Actinobacteria/Actinomycetota being the most abundant ones. Although a correlation was not found between these phyla and th HIV status, a trend was observed toward a higher relative abundance of Bacteroidetes/Bacteroidota within the control group (*p* = 0.072; Table 2).

Overall, a total of 135 bacterial genera were detected in this study. Some significant differences were found between both groups: the relative abundance of the genus *Blautia* was higher in the PWH group (*p* < 0.001), whereas the opposite was found for the genera *Bacteroides*, *Oscillospira*, and *Clostridium* (*p* = 0.033, *p* = 0.018, and *p* = 0.005, respectively; Table 2; Figure 2).

When the sequences from the three PWH subgroups (i.e., frail, robust, and prefrail patients) were compared, no differences were found in relation to alpha and beta diversity or the taxonomic composition at the phyla and genera levels (Table 3).

Relative abundance of *Blautia* was higher among those PWH patients with depression (5.23 [13.96–16.99] vs. 10.38 [8.87–11.5]: *p* = 0.004), whereas the opposite was observed for the genus *Bifidobacterium* (3.07 [0.2–11.58] vs. <0.01 [<0.01–0.37], *p* = 0.022). No differences were found among the patients of the PWH group when they were compared depending on frailty status, current and nadir CD4 status, CD4/CD8 ratio, years of HIV infection, body mass index, polypharmacy, and the remaining basal characteristics (Appendix A).

## 4. Discussion

In this study, we observed significant differences between the fecal microbiota of the PWH and control groups. This finding is congruent with those reported by other authors in previous studies, which showed that people living with HIV present alterations in the composition of their fecal microbiotas that are similar to those caused by aging [20,27,28], including decreased alpha diversity and lower abundance [29], which may be positively correlated with systemic inflammatory markers [19,30,31,32]. Correlations between gut dysbiosis, inflammation, and an increase in circulating biomarkers of gut epithelial barrier damage and microbial translocation have been found in a nonhuman primate model of HIV infection [33]. These authors suggested that gut dysbiosis may maintain inflammation and metabolic alterations in chronic HIV patients despite proper long-term control of viremia.

The most abundant phyla identified in our study were Firmicutes/Bacillota, Bacteroidetes/Bacteroidota, Proteobacteria/Pseudomonadota, and Actinobacteria/Actinomycetota. HIV did not have a significant effect on the relative abundance of these phyla; however, the trend toward a lower abundance of the phylum Bacteroidetes in the PWH group is congruent with previously published data [28,34,35,36]. At the genus level, we detected 135 genera, and there were statistically significant differences between both groups in relation to the relative abundances of the genus *Blautia*, which was higher in the PWH group. *Blautia* is a commensal bacterial species that can become pathogenic, and, in fact, a higher abundance of this genus has been associated with some conditions, including irritable bowel syndrome, ulcerative colitis, and early breast cancer [37,38,39]. Previous studies have addressed the potential association between *Blautia* and HIV patients, and the results appear to be somehow controversial. So, while one study found that *Blautia* was highly relevant in HIV-infected individuals [40], another one found that this genus was enriched in naive HIV-infected patients but depleted in HIV-positive elite controllers and in HIV-negative individuals [41]. More recently, it has been described that *Blautia* sequences were more abundant in the colon and ileal samples collected by colonoscopy from HIV patients on antiretroviral therapy than in those from HIV-negative patients [42]. 

In addition, our results showed that the relative abundance of *Blautia* was higher in those PHW patients diagnosed with depression, whereas that of *Bifidobacterium* was higher in those who did not have depression. Although the etiopathogenesis of depression and other psychiatric disorders is still not completely elucidated, recent findings suggest that the dysbiosis of the gut microbiota might play a role in the severity of symptoms and, also, in modulating the efficacy and safety of treatments [43,44]. The relationship between depressive disorders and a higher abundance of *Blautia* and a lower one of *Bifidobacterium* in the fecal microbiome has already been reported [45,46,47,48,49], but, to our knowledge, this is the first study in which such potential associations have been linked to an HIV population. In other studies, changes in the fecal microbiome during the development of depressive-like behaviors in rats exposed to chronic unpredictable mild stress (CUMS) were assessed and compared with healthy controls [50]. Interestingly, the genus *Blautia* was reported to be more abundant in the CUMS group than in the healthy group. Interestingly, a higher abundance of *Blautia* has also been reported among HIV patients reporting distal neuropathic pain [29]. Although our study does not allow us to establish causality between depression and the presence of *Blautia* in HIV patients, this possibility should be the subject of future investigations.

On the other hand, we found that the abundance of the genera *Bacteroides* and *Oscillospira* was higher in the healthy control group. These findings are also consistent with previously published studies that have found significantly fewer *Bacteroides* in the PWH group than in the uninfected controls [28,30,31,32]. The decrease in the fecal abundance of some *Bacteroides* species, including *B. ovatus* and *B. thetaiotaomicron*, seems to be a feature of aging [51]. A significant reduction in the percentage of sequences belonging to this genus has also been detected in the context of other viral diseases, such as COVID-19, in patients when compared to healthy controls [52]. This fact may render the elderly population more vulnerable to COVID-19 since *Bacteroides* spp. have been associated with antiviral activity through a variety of mechanisms, including the downregulation of the expression of viral receptors and the alteration of receptor binding by their heparan sulfate-modifying glycosidase activities [53]. *Oscillospira* is a genus of commensal bacteria that produces butyrate and is commonly found in the gut of healthy hosts. Its abundance is negatively associated with metabolic syndrome-related parameters and with other diseases that involve inflammation [54,55]. However, the relationship between *Oscillopira* and HIV infection seems controversial since one report showed an abundance of this genus among HIV-positive women in comparison with healthy controls [56], while others revealed that this genus was more abundant in HIV-positive elite controllers than in naive HIV-infected patients [41].

Finally, another relevant finding of our study is the absence of differences in the composition of the microbiota when the different subgroups of HIV-infected patients were compared according to their frailty status (i.e., frail, robust, or pre-frail). So far, data from the literature focused on the possible relationship between frailty and dysbiosis are inconclusive. While some studies have observed a lower abundance of butyrate-producing bacteria in frail patients, scientific evidence is limited regarding microbiome-related biomarkers, enabling a clear differentiation between frail and non-frail patients [57,58,59,60,61].

The main limitation of our study is the small sample size, which may be one of the reasons we did not find differences between the frail PWH and control groups or between polypharmacy and non-polypharmacy patients. Multicenter studies with larger cohorts are required to obtain more conclusive data in this field. Another limitation could be the fact that we do not have an in-depth analysis of all the foods in the patients’ diet. However, there were no baseline differences, including years of HIV or BMI; none of the participants had received antibiotics in the previous 3 months; none of the participants had major dietary restrictions (e.g., vegan or vegetarian); and all were matched for age and sex at birth and came from the same geographical area.

## 5. Conclusions

In conclusion, our study further suggests the existence of an alteration of the fecal bacteriome in older adults with HIV when compared to healthy controls, a fact that may lead to the development of future strategies for modulating the gut microbiome of HIV patients. The altered gut bacteriome associated with HIV infection is not restored with correct antiretroviral therapy [62]. This fact may affect therapy outcomes since gut microbes and their metabolites play key roles in the development and regulation of host immunity [63]. As stated by Li et al. [62], understanding the factors shaping the so-called HIV-associated microbiome seems critical for developing novel approaches and therapies to improve the health of HIV patients.

## Figures and Tables

**Figure 1 biomedicines-11-02305-f001:**
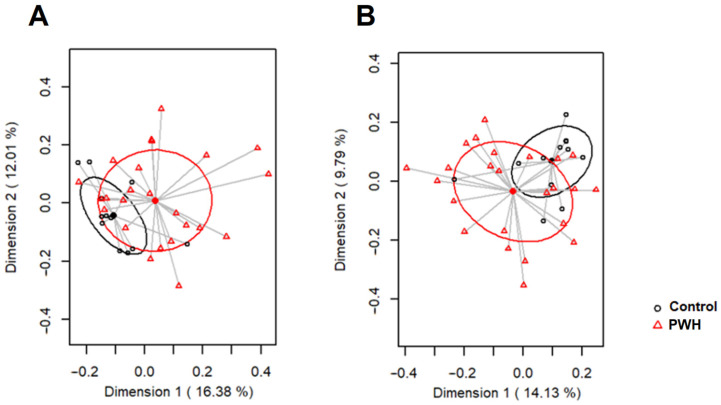
PCoA plots of bacterial profiles (at the OTUs level), based on the Bray–Curtis similarity analysis (**A**) and Jaccard’s coefficient for binary data (**B**) from the fecal samples of control patients (control, black circles) and PWH patients (red triangles). The percentage of the total variance is explained by each axis.

**Figure 2 biomedicines-11-02305-f002:**
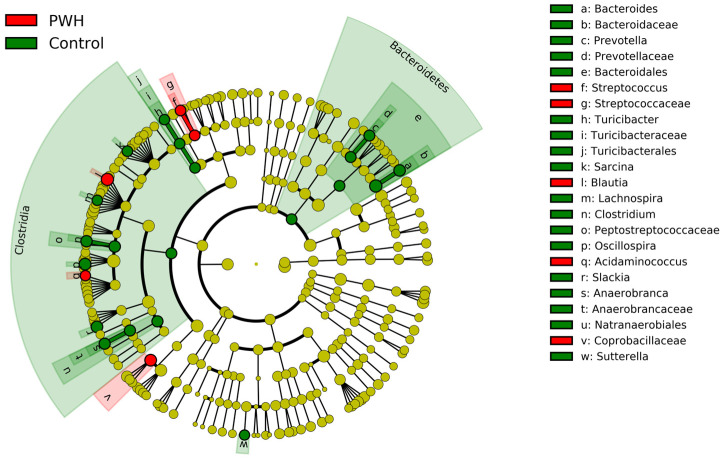
LEfSe comparison between the abundance relative bacterial profile detected in the two groups (PWH [red] and control [green]) analyzed in this study.

**Table 1 biomedicines-11-02305-t001:** Main characteristics of the PWH patients and controls included in this study.

	All PWH N = 24	Frail-PWH N = 7	Non-Frail-PWH N = 17		Healthy Controls N = 12	*p*
			Prefrail N = 9	Robust N = 8		
Sex at birth (Women/Men)	**4 W/20 M**	**2 W/5 M**	**2 W/7 M**	0 W/8 M	4 W/8 M	
Age (years) Media (SD)	61.9 (7.6)	60 (3.9)	65.3 (10.9)	59.7 (4.2)	60.6 (6.1)	0.245
Years living with known HIV Media (SD)	16.5 (7.7)	19.8 (6.5)	17.5 (8.5)	12.5 (6.6)	-	0.164
CD4+ nadir cells/mm^3^ Median (IQR)	170.6 (154)	171 (141.2)	116.6 (105.6)	231 (200)	-	0.325
Current CD4+ (cells/mm^3^) Media (SD)	589.2 (342.1)	662.8 (579.4)	523.7 (207.8)	598.5 (191)	-	0.737
Rate CD4/CD8 Median (IQR)	0.87 (0.5)	0.70 (0.3)	0.74 (0.3)	1.1 (0.6)	-	0.124
Diagnosed with depression (N)	7	3	2	2	-	0.634
Polypharmacy * (N)	12	6	5	1	-	0.017
BMI kg/m^2^ (N)					-	0.293
<25	15	5	6	4		
25–29	5	0	3	2		
>29	4	2	0	2		

* Polypharmacy was defined as taking at least six or more comedications (excluding antiretroviral treatment).

**Table 2 biomedicines-11-02305-t002:** Relative abundance (%), expressed as median and interquartile ranges, of the most abundant genera and phyla detected in the two groups (PWH and control) analyzed in this study.

	Control		PWH		
Phyla/Genera	N (%) ^a^	Median (IQR)	N (%)	Median (IQR)	*p*-Value *
Firmicutes/Bacillota	12 (100%)	73.27 (67.71–77.6)	24 (100%)	74.51 (66.49–80.73)	0.730
* Faecalibacterium *	12 (100%)	9.25 (8.4–14.04)	24 (100%)	9.1 (5.07–15.04)	0.560
* Blautia *	12 (100%)	7.16 (5.82–8.48)	24 (100%)	11.18 (9.53–14.57)	<0.001
* Ruminococcus *	12 (100%)	9.99 (8.12–14.64)	24 (100%)	9.08 (6.42–11.71)	0.500
* Clostridium *	12 (100%)	5.49 (4.17–7.21)	24 (100%)	3.08 (2.42–3.72)	0.005
* Collinsella *	10 (83.33%)	2.59 (1.11–4.78)	21 (87.5%)	2.83 (0.67–7.56)	0.450
* Coprococcus *	12 (100%)	1.79 (0.79–3.13)	22 (91.67%)	2.54 (1.32–4.58)	0.180
* Slackia *	12 (100%)	3.23 (1.45–3.93)	23 (95.83%)	2.01 (0.38–3.88)	0.180
* Oscillospira *	12 (100%)	3.99 (2.5–4.49)	23 (95.83%)	1.32 (0.78–2.89)	0.018
* Alkaliphilus *	11 (91.67%)	3.44 (2.08–4.46)	22 (91.67%)	0.76 (0.33–2.13)	0.010
* Catenibacterium *	2 (16.67%)	<0.01 (<0.01–<0.01)	9 (37.5%)	<0.01 (<0.01–1.83)	0.210
* Roseburia *	12 (100%)	1.54 (0.86–2.49)	22 (91.67%)	1.21 (0.53–2.59)	0.700
* Eubacterium *	9 (75%)	0.57 (0.06–2.23)	17 (70.83%)	0.64 (<0.01–2.48)	0.970
* Erysipelothrix *	10 (83.33%)	0.55 (0.25–1.4)	24 (100%)	0.83 (0.39–1.65)	0.250
* Dorea *	12 (100%)	0.58 (0.35–0.71)	23 (95.83%)	0.9 (0.66–1.38)	0.022
Bacteroidetes/Bacteroidota	12 (100%)	10.11 (6.6–13.5)	23 (95.83%)	7.31 (1.86–9.81)	0.072
* Bacteroides *	12 (100%)	5.89 (4.36–8.97)	21 (87.5%)	1.54 (0.5–6.61)	0.033
Proteobacteria/ Pseudomonota	12 (100%)	1.89 (1.45–5.5)	24 (100%)	1.55 (0.88–4.79)	0.560
* Escherichia *	8 (66.67%)	0.37 (<0.01–1.22)	13 (54.17%)	0.18 (<0.01–1.92)	0.740
Actinobacteria/ Actinomycetota	12 (100%)	3.33 (1.36–6.68)	19 (79.17%)	1.29 (0.39–4.41)	0.320
* Bifidobacterium *	12 (100%)	3.21 (0.63–6.55)	15 (62.5%)	0.99 (<0.01–4.17)	0.240
Minor_phyla	9 (75%)	0.55 (<0.01–0.88)	20 (83.33%)	0.85 (0.3–2.23)	0.310
* Akkermansia *	4 (33.33%)	<0.01 (<0.01–0.25)	7 (29.17%)	<0.01 (<0.01–0.26)	0.950
Minor_genera	12 (100%)	13.87 (10.3–17.63)	24 (100%)	11.99 (8.66–19.25)	0.750
Unclassified_phyla	12 (100%)	7.21 (6.66–7.58)	24 (100%)	7.09 (6.46–7.66)	0.700
Unclassified_genera	12 (100%)	15.01 (13.96–15.5)	24 (100%)	12.22 (11.16–14.94)	0.210

^a^: Number of samples in which the phylum/genus was detected (relative frequency of detection). * Wilcoxon rank test.

**Table 3 biomedicines-11-02305-t003:** Alpha and beta diversity and relative abundance (%), expressed as median and interquartile ranges, of the most abundant genera and phyla detected in the three PWH subgroups (robust, pre-frail, and frail) analyzed in this study.

	Robust (n = 8)		Pre-Frail (n = 9)		Frail (n = 7)		*p*-Value *
Shannon index	3.38 (3.28–3.65)		3.63 (3.51–3.7)		3.59 (3.37–3.62)		0.45
Bray–Curtis ^a^	A		A		A		0.70 **
Jaccard ^a^	A		A		A		0.19 **
Phylum/genera	N (%)	median (IQR)	N (%)	median (IQR)	N (%)	median (IQR)	
Firmicutes/Bacillota	8 (100%)	75.48 (66.12–82.3)	9 (100%)	70.64 (64.59–76.33)	7 (100%)	78.84 (69.33–81.75)	0.79
*Blautia*	8 (100%)	10.22 (8.31–12.12)	9 (100%)	10.81 (9.11–13.66)	7 (100%)	13.41 (11.39–15)	0.23
*Faecalibacterium*	8 (100%)	8.84 (3.43–12.44)	9 (100%)	10.59 (8.5–18.26)	7 (100%)	7.12 (4.12–15.45)	0.47
*Ruminococcus*	8 (100%)	9.08 (8.49–12.74)	9 (100%)	11.12 (4.77–11.7)	7 (100%)	8.7 (6.87–11.76)	0.88
*Collinsella*	7 (87.5%)	4.56 (0.63–10.24)	8 (88.89%)	2.69 (2.3–3.61)	6 (85.71%)	5.14 (1.61–7.71)	0.74
*Clostridium*	8 (100%)	2.96 (2.33–4)	9 (100%)	3.22 (2.98–3.63)	7 (100%)	2.77 (1.59–3.67)	0.47
*Coprococcus*	8 (100%)	1.96 (1.35–3.22)	9 (100%)	3.79 (2.27–4.08)	5 (71.43%)	2.22 (0.67–5.09)	0.60
*Slackia*	7 (87.5%)	2.57 (1.16–3.9)	9 (100%)	0.97 (0.46–2.88)	7 (100%)	2.12 (0.34–4.48)	0.79
*Catenibacterium*	4 (50%)	0.27 (<0.01–4.37)	2 (22.22%)	<0.01 (<0.01–<0.01)	3 (42.86%)	<0.01 (<0.01–1.89)	0.41
*Oscillospira*	8 (100%)	1.41 (0.77–2.37)	9 (100%)	1.59 (0.8–4.21)	6 (85.71%)	1.05 (0.8–2.49)	0.87
*Roseburia*	8 (100%)	0.84 (0.6–1.42)	9 (100%)	1.86 (1.29–3.27)	5 (71.43%)	1.14 (0.27–3.25)	0.56
*Eubacterium*	5 (62.5%)	1.51 (<0.01–4.01)	6 (66.67%)	0.27 (<0.01–1.3)	6 (85.71%)	1.3 (0.13–2.3)	0.71
*Erysipelothrix*	8 (100%)	0.8 (0.47–2.18)	9 (100%)	0.48 (0.36–0.78)	7 (100%)	1.45 (0.9–2.64)	0.13
*Alkaliphilus*	7 (87.5%)	0.42 (0.2–0.91)	9 (100%)	0.94 (0.58–1.52)	6 (85.71%)	1.58 (0.55–2.88)	0.22
*Dorea*	8 (100%)	1.4 (1.02–1.74)	9 (100%)	0.77 (0.64–0.85)	6 (85.71%)	0.89 (0.45–1.58)	0.08
Bacteroidetes/ Bacteroidota	7 (87.5%)	5.84 (0.72–10.17)	9 (100%)	7.43 (5.96–8.95)	7 (100%)	2.91 (1.17–8.26)	0.42
*Bacteroides*	6 (75%)	1.05 (0.39–3.22)	9 (100%)	3.76 (1.55–6.59)	6 (85.71%)	0.44 (0.2–4.19)	0.16
Proteobacteria/ Pseudomonodota	8 (100%)	1.55 (0.91–3.96)	9 (100%)	4.4 (1.32–9.24)	7 (100%)	1.42 (0.85–2.5)	0.38
*Escherichia*	4 (50%)	0.12 (<0.01–0.3)	5 (55.56%)	1.84 (<0.01–5.31)	4 (57.14%)	0.11 (<0.01–0.75)	0.62
Actinobacteria/ Actinomycetota	5 (62.5%)	0.74 (<0.01–6.77)	8 (88.89%)	1.23 (0.63–3.4)	6 (85.71%)	3.74 (0.96–4.04)	0.79
*Bifidobacterium*	3 (37.5%)	<0.01 (<0.01–6.26)	6 (66.67%)	1.23 (<0.01–3.07)	6 (85.71%)	3.01 (0.47–3.72)	0.68
Minor_phyla	7 (87.5%)	0.68 (0.24–1.08)	7 (77.78%)	1 (0.43–2.22)	6 (85.71%)	0.72 (0.33–10.34)	0.79
*Akkermansia*	2 (25%)	<0.01 (<0.01–0.05)	2 (22.22%)	<0.01 (<0.01–<0.01)	3 (42.86%)	<0.01 (<0.01–8.1)	0.51
Minor_genera	8 (100%)	10.23 (7.57–14.83)	9 (100%)	11.91 (10.59–19.03)	7 (100%)	12.07 (9.1–18.79)	0.68
Unclassified_phyla	8 (100%)	7.33 (6.56–7.74)	9 (100%)	6.77 (6.45–7.17)	7 (100%)	7.26 (6.73–7.71)	0.46
Unclassified_genera	8 (100%)	12.38 (11.77–15.71)	9 (100%)	11.85 (11.21–13.19)	7 (100%)	12.33 (11.15–13.89)	0.88

^a^: No significant differences were observed among groups displaying the same letter. * Kruskal–Wallis test. ** PERMANOVA test with 999 permutations.

## Data Availability

The data presented in this study are available on request from the corresponding author.

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
