# Peer review of "Comparison of the Fecal Bacteriome of HIV-Positive and HIV-Negative Older Adults"

_biomedicines, 2023, doi:10.3390/biomedicines11082305_

Round 1

Reviewer 1 Report

Dear authors

This manuscript has evaluated the association of gut microbiota and its diversity with HIV and frailty and weakness, and depression.

There are following comments for this manuscript:

1-      There are a few similar previous studies. The authors need to use these studies in the introduction or discussion:

Unique gut microbiome in HIV patients on antiretroviral therapy (ART) suggests association with chronic inflammation. Microbiology Spectrum

Gut microbiota linked to sexual preference and HIV infection. EBioMedicine. 2016

The gut microbiome and HIV-1 pathogenesis: a two way street. AIDS (London, England). 2016

Complexities of gut microbiome dysbiosis in the context of HIV infection and antiretroviral therapy

Evolution of the Gut Microbiome in HIV-Exposed Uninfected and Unexposed Infants during the First Year of Life. Mbio. 2022

Reduced gut microbiome diversity in people with HIV who have distal neuropathic pain. The journal of pain. 2022

Antenatal gut microbiome profiles and effect on pregnancy outcome in HIV infected and HIV uninfected women in a resource limited setting. BMC microbiology. 2023

Gut microbiome changes associated with epithelial barrier damage and systemic inflammation during antiretroviral therapy of chronic SIV infection. Viruses. 2021

The gut microbiome and type 2 diabetes mellitus: discussing a complex relationship. Biomedicines. 2020

Gastrointestinal microbiome and neurologic injury. Biomedicines. 2022

Involvement of gut microbiota in schizophrenia and treatment resistance to antipsychotics. Biomedicines. 2021

Exploring the role of gut microbiota in major depressive disorder and in treatment resistance to antidepressants. Biomedicines. 2020

Crosstalk between gut microbiota and host immunity: Impact on inflammation and immunotherapy. Biomedicines. 2023

The Gut Microbiome–Brain Crosstalk in Neurodegenerative Diseases. Biomedicines. 2022

Suppressive Effects of Lactobacillus on Depression through Regulating the Gut Microbiota and Metabolites in C57BL/6J Mice Induced by Ampicillin. Biomedicines. 2023.  

2-      The title and abstract of the manuscript are appropriate and relative to the text. However, more details of methods can be added in the abstract. The p values of main findings can be added in the abstract.

3-      The introduction section can be improved in the aspect of risk factors. There are more studies to be discussed as stated in the first comment.

4-      Methods section is suitable. However, if there are any references, they can be added to this section.

5-      Please add p-value scores in addition to percentages in parentheses.  

6-      The writing of the manuscript is appropriate and needs minor revision.

7-      Results are appropriate and the findings are valuable.

8-       Please follow the journal style throughout the manuscript.  

9-      Please use more related studies from 2022 in the discussion section.

10-   Please also use this reference:

Microbiome Dysbiosis and Predominant Bacterial Species as Human Cancer Biomarkers

With best regards 

Dear editor 

The writing is suitable and minor revisions are needed. 

Best regards 

Author Response

Thank you very much for your comments, which are very helpful for enhancing our work. Below, we provide a more detailed response to each of them:

Q1. There are a few similar previous studies. The authors need to use these studies in the introduction or discussion:

Answer: Thank you for the comment and suggestions. We agree with you that there are a few studies in the field of dysbiosis and HIV infection, although there are not many that are specifically focused on the older adult population with HIV. We appreciate your suggestions very much and we have incorporated most of them in the introduction or discussion sections of the revised manuscript and, of course, in the references list. The ones that we have included (in order of appearance in the text), are the following:

[2] Dillon SM, Frank DN, Wilson CC. The gut microbiome and HIV-1 pathogenesis: a two-way street. AIDS. 2016 Nov 28;30(18):2737-2751. doi: 10.1097/QAD.0000000000001289.

[3] Ishizaka A, Koga M, Mizutani T, Parbie PK, Prawisuda D, Yusa N, Sedohara A, Kikuchi T, Ikeuchi K, Adachi E, Koibuchi T, Furukawa Y, Tojo A, Imoto S, Suzuki Y, Tsutsumi T, Kiyono H, Matano T, Yotsuyanagi H. Unique Gut Microbiome in HIV Patients on Antiretroviral Therapy (ART) Suggests Association with Chronic Inflammation. Microbiol Spectr. 2021 Sep 3;9(1):e0070821. doi: 10.1128/Spectrum.00708-21. 

[29] Ellis RJ, Heaton RK, Gianella S, Rahman G, Knight R. Reduced Gut Microbiome Diversity in People With HIV Who Have Distal Neuropathic Pain. J Pain. 2022 Feb;23(2):318-325. doi: 10.1016/j.jpain.2021.08.006. 

[33] Tanes C, Walker EM, Slisarenko N, Gerrets GL, Grasperge BF, Qin X, Jazwinski SM, Bushman FD, Bittinger K, Rout N. Gut Microbiome Changes Associated with Epithelial Barrier Damage and Systemic Inflammation during Antiretroviral Therapy of Chronic SIV Infection. Viruses. 2021 Aug 8;13(8):1567. doi: 10.3390/v13081567. 

[43] Fontana A, Manchia M, Panebianco C, Paribello P, Arzedi C, Cossu E, Garzilli M, Montis MA, Mura A, Pisanu C, Congiu D, Copetti M, Pinna F, Carpiniello B, Squassina A, Pazienza V. Exploring the Role of Gut Microbiota in Major Depressive Disorder and in Treatment Resistance to Antidepressants. Biomedicines. 2020 Aug 27;8(9):311. doi: 10.3390/biomedicines8090311. 

[44] Manchia M, Fontana A, Panebianco C, Paribello P, Arzedi C, Cossu E, Garzilli M, Montis MA, Mura A, Pisanu C, Congiu D, Copetti M, Pinna F, Pazienza V, Squassina A, Carpiniello B. Involvement of Gut Microbiota in Schizophrenia and Treatment Resistance to Antipsychotics. Biomedicines. 2021 Jul 23;9(8):875. doi: 10.3390/biomedicines9080875. 

[62] Li SX, Armstrong A, Neff CP, Shaffer M, Lozupone CA, Palmer BE. Complexities of Gut Microbiome Dysbiosis in the Context of HIV Infection and Antiretroviral Therapy. Clin Pharmacol Ther. 2016 Jun;99(6):600-11. doi: 10.1002/cpt.363. 

[63] Campbell C, Kandalgaonkar MR, Golonka RM, Yeoh BS, Vijay-Kumar M, Saha P. Crosstalk between Gut Microbiota and Host Immunity: Impact on Inflammation and Immunotherapy. Biomedicines. 2023 Jan 20;11(2):294. doi: 10.3390/biomedicines11020294. 

In addition, we have included two new references since we think they are relevant for this specific topic, and they had not been included in the original manuscript:

[5] Zevin AS, McKinnon L, Burgener A, Klatt NR. Microbial translocation and microbiome dysbiosis in HIV-associated immune activation. Curr Opin HIV AIDS. 2016 Mar;11(2):182-90. doi: 10.1097/COH.0000000000000234. 

[42] Meng J, Tao J, Abu Y, Sussman DA, Girotra M, Franceschi D, Roy S. HIV-Positive Patients on Antiretroviral Therapy Have an Altered Mucosal Intestinal but Not Oral Microbiome. Microbiol Spectr. 2023 Feb 14;11(1):e0247222. doi: 10.1128/spectrum.02472-22. 

Q2. The title and abstract of the manuscript are appropriate and relative to the text. However, more details of methods can be added in the abstract. The values of main findings can be added in the abstract.

Answer: According to your suggestion, we have included additional information regarding the methods and p values of the main findings in the abstract, while keeping it in a maximum of 200 words.

Q3. The introduction section can be improved in the aspect of risk factors. There are more studies to be discussed as stated in the first comment.

Answer: Thank you very much. We have incorporated some of the references suggested by the reviewer (in Q1) in the introduction of the revised manuscript.

Q4. Methods section is suitable. However, if there are any references, they can be added to this section.

Answer.All the references required for a full description of the methods used in this study have been included in the revised manuscript.

Q5. Please add p-value scores in addition to percentages in parentheses.  

Answer: P-values are included in addition to percentages in parentheses.

Q6. The writing of the manuscript is appropriate and needs minor revision.

A6. Thank you for your comment. We have performed an extensive review of the English usage of this manuscript and minor errors and mistakes have been corrected.

Q7. Results are appropriate and the findings are valuable.

Answer:Thank you very much for your comment.

Q8. Please follow the journal style throughout the manuscript.  

Answer: The journal style have been followed throughout the manuscript (including the references).

Q9. Please use more related studies from 2022 in the discussion section.

Answer: Thank you very much. We have incorporated most of the references suggested by the reviewer (in Q1), together with a few additional ones published between 2021 and 2023, in the discussion of the revised manuscript.

Q10. Please also use this reference: Microbiome Dysbiosis and Predominant Bacterial Species as Human Cancer Biomarkers

Answer:This reference has been also included in the introduction section of the revised manuscript.

[4] Shirazi, M.S.R.; Al-Alo, K.Z.K.; Al-Yasiri, M.H.; Lateef, Z.M.; Ghasemian, A. microbiome dysbiosis and predominant bacterial species as human cancer biomarkers. J. Gastrointest. Cancer 2020, 51, 725-728.

Reviewer 2 Report

The authors present a concise manuscript comparing the fecal bacteriome among PWH and non-PWH.  The authors attempt to make a connection between an increase in frailty with a concomitant increase in gut dysbiosis among PWH to non-PWH. Although, this connection was not observed, the authors clearly present some differences in the fecal bacteriome in PWH and non-PWH. However, a novel finding revealed a connection between HIV-1 and depression based off the relative abundances of certain fecal microbiome genera was made and warrants future studies.

First paragraph of the discussion needs to be removed. Lines 189-192.

Author Response

Thank you for your kind review. As suggested, we have removed the first paragraph of the discussion.